# Invariant Feature Learning by Attribute Perception Matching

**Yusuke Iwasawa, Kei Akuzawao & Yutaka Matsuo**
Department of Technology Management for Innovation, The University of Tokyo
`{iwasawa,akuzawa-kei,matsuo}@weblab.t.u-tokyo.ac.jp`

## Abstract

An adversarial feature learning (AFL) is a powerful framework to learn representations invariant to a nuisance attribute, which uses an adversarial game between a feature extractor and a categorical attribute classifier. It theoretically sounds in term of it maximize conditional entropy between attribute and representation. However, as shown in this paper, the AFL often causes unstable behavior that slows down the convergence. We propose an *attribute perception matching* as an alternative approach, based on the reformulation of conditional entropy maximization as *pair-wise distribution matching*. Although the naive approach for realizing the pair-wise distribution matching requires the significantly large number of parameters, the proposed method requires the same number of parameters with AFL but has a better convergence property. Experiments on both toy and real-world dataset prove that our proposed method converges to better invariant representation significantly faster than AFL.

## 1 Introduction

How to learn representations invariant to nuisance attribute $a$ is technical challenges raised in domain generalizaton (Blanchard et al., 2011; Muandet et al., 2013; Ghifary et al., 2015; Motiian et al., 2017), fair classification, privacy-protection (Edwards & Storkey, 2016; Iwasawa et al., 2017), and many other area. Assume that we are given a training dataset made of pairs $\mathcal{S} = \left\{ (x_i, y_i, a_i) \right\}_{i=1}^{n}$, where $x$ is an observation, $y$ is a target of $x$, and $a$ is a corresponding intrinsic attribute of $K$-way categorical variable $A$. The goal of invariant representation learning is to obtain an encoder $E$ that reduces information about attribute $a$ while maintaining information about $y$.

An adversarial game between a feature extractor and an attribute classifier, called *adversarial feature learning* (Xie et al., 2017), is a powerful framework for this purpose. The key of AFL is to measure the invariance by leveraging the discriminative power of neural network beyond the pre-defined metric such as $l_2$ distance or maximum mean discrepancy. That is, if the external network (also referred to as a discriminator) can predict $a$ from $z = E(x)$, AFL regards $z$ to have considerable information about $a$. Formally, the AFL solves the following optimization problem:

$$\min_{E,M} \max_{D} \mathop{\mathbb{E}}_{x,y,a \in \mathcal{S}} \left[ -\log q_M(y|E(x)) + \lambda \log q_D(a|E(x)) \right], \tag{1}$$

where $q_M$ and $q_D$ is the conditional probability that $M$ and $D$ gives a correct estimation of $y$ and $a$ respectively. As Xie et al. (2017) explained, this alternating procedure can be regarded as a way to maximize the conditional entropy $H(A|Z) = \sum_{a \in \mathcal{A}, z \in \mathcal{Z}} -p(a, z) \log p(a|z)$, where $\mathcal{A}$ and $\mathcal{Z}$ is a support set of $a$ and $z$. Xie et al. (2017) also showed that the min–max game has an equilibrium, in which $E$ maximize the conditional entropy $H(A|Z)$. It has been show superior performance in fair-classification, privacy-protection, and domain generalization tasks (Ganin et al., 2016; Edwards & Storkey, 2016; Xie et al., 2017; Iwasawa et al., 2017), compared to the predifined metric approaches (Zemel et al., 2013; Louizos et al., 2016; Motiian et al., 2017).

Despite the theoretical justifications, the above min–max formulation is suspicious for several practical issues. Namely, the gradient from the discriminator vanishes if the discriminator sufficiently trained since $\mathbb{E}[\log q_D(a|z{=}E(x))]$ is small then. Besides, in mathematical level, it only keeps away representations from the non-desired point where we can predict a label correctly, but not ensure that it approaches the desired invariant point. Please also refer Fig. 1 for visualization of the instability.

Note that, Generative Adversarial Networks community, which utilize similar formulation to generate realistic images, evade similar issues by the incorporating alternative objectives, such as the Wasserstein distance (Arjovsky et al., 2017). However, the Wasserstein distance is defined over two distributions and applying to our setting (consisting of multiple distributions) is not trivial.

This paper holds the following contributions to the invariant feature learning problem. First, we empirically show that AFL is suffered from practical issues that significantly slow down the convergence. We then reformulate the optimization problem of AFL as pair-wise distribution matching and derive parameter practical realization of pairwise distribution matching while inheriting the merit of AFL that leveraging the discriminative power to measure the invariance. It is worth mentioning that the reformulation enable us to use Wasserstein metric in theory, however, it is still computationally infeasible in practice because a naive way to calculate the Wasserstein distance between all the pair of the distributions requires $O(K^2)$ discriminators, where $K = |\mathcal{A}|$, which raise computational issues both in terms of parameter size and forward/backward time. Finally, we empirically validate the superior performance of our proposed method on both artificial dataset and real-world datasets.

## 2 CONVERGENCE ISSUES OF AFL

Figure 1-(a–e) visualize a behavior of AFL optimization on synthesized data. Each figure corresponds to the different timestep of the alternating optimization. The dataset consists of samples from three Gaussian distributions with different means ($[\sin(\frac{i}{3}\pi), \cos(\frac{i}{3}\pi)]$, for $i \in 1, 2, 3$, respectively) and the same variance, assuming that each distribution corresponds to different attributes. In each figure, dots represent the data point, color represents the attribute (domain id), and the contour plot represents the discriminator's decision boundary. A float value on the top of figures is the negative log-likelihood (NLL) of the dataset measured by the discriminator $D$ (the multi-layer perceptron with 100 hidden units followed by a ReLU activation). Similarly, a float value in parenthesis on the top of figures is an NLL of a post-hoc classifier $D_{eval}$ that have the same architecture as $D$. To be more specific, we first train the discriminator 100 times with 128 batch size and train $D$ and $E$ iteratively with stochastic gradient descent with learning rate=0.1. Figure 1-(f,g) shows the gradient vector fields of different time steps for $a = blue$, where the arrow represents the direction of the gradient, and the norm represents its magnitude. For simplicity, we only show the vector fields of $a = blue$, but the observations are quite similar for the other $a$.

The figure reveals two practical issues in AFL optimization. (1) The distribution alignment is initially quite slow (compare with the behavior of the proposed method shown in Figure 2). This is because the gradient is small when the discriminator correctly distinguishes a $a$. (2) AFL behavior is unstable. The distributions somewhat align after 40 steps (given 0.683 NLL with the post-hoc classifier), but it is catastrophically failed five steps later because the discriminator did not capture the true conditional entropy (implied by the mostly similar counterplot of $D$) and therefore gave a false gradient as shown in (f) and (g). The intuitive reason for this phenomenon is that AFLs loss essentially tries to pull a distribution apart from the non-desired point, i.e., the point where we can correctly predict the label. The problem of AFL is that it only keeps away a distribution from the non-desired point, but not ensure it approaches the desired invariant point. After several steps, $D$ starts to follow the change of the distribution (as shown in Figure 1-e). The instability of the AFL also appears in the significant gap between the NLL of the $D$ and $D_{eval}$. Note that the second issue may be alleviated if $D$ has a sufficiently large capacity and is trained many times at each iteration. However, this is not a realistic assumption since it is fair to say that real datasets are more complicated than this toy situations, making it more challenging to find the supremum.

## 3 EFFICIENT PAIWRISE DISTRIBUTION MATCHING

The next question we must answer is how to maximize conditional entropy while avoiding the issues mentioned above. Assume that $A$ is a categorical random variable drawn from a uniform discrete distribution, and denote that support set of $a$ and $z$ as $\mathcal{A}$ and $\mathcal{Z}$; then, the following theorem holds:

**Theorem 1.** *The maximum conditional entropy $H(A|Z)$ is $-\log \frac{1}{K}$, and $H(A|Z)$ is maximized if and only if $p(z|a{=}i) = p(z|a{=}j)$ for all $i \neq j \in 1, \cdots, K$ and $z \in \mathcal{Z}$.*

The proof is followed in the appendix. One implication of the theorem is, the empirical measurements of conditional entropy should be bounded. It suggests another problematic point of AFL objective since it is not lower bounded. Another implication is that this theorem permits us to rethink the problem of the conditional entropy maximization as a problem of aligning all pairs of

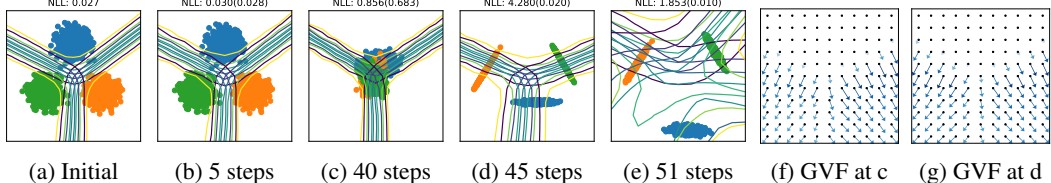

Figure 1: Visualization of the unstable behavior of AFL. (a) A dataset used for the experiments. (b–e) The visualization of representation space at different timesteps. (f, g) The gradient vector fields (GVF) for blue data-points at (f) 38 steps and (g) 41 steps.

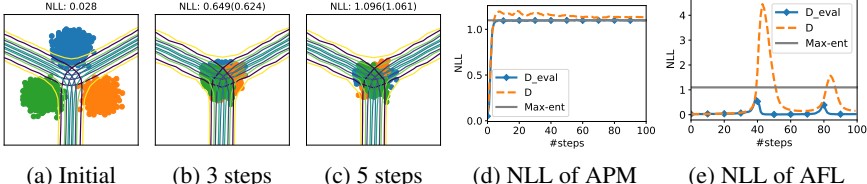

Figure 2: Visulization of the behavior of the proposed method in the toy dataset.

distributions $p(z|a = i)$ and $p(z|a = j)$. This reformulation is significant since we can now use any conventional measurement of two distributions. However, the naive approach requires to match all pairs of the distributions ($=_K C_2$), which would be cumbersome to compute.

We, therefore, propose a computationally efficient way to realize pairwise distribution matching, *moving attribute perception matching*. As with AFL, APM is based on the alternating training of attribute classifier $D$ and feature extractor $E$, but APM deceives $D$ differently. Formally, we propose the following APM matching objective:

$$V_{map}(E, D) = \underset{x,y,a \in \mathcal{S}}{\mathbb{E}} \Big[ \frac{1}{K-1} \sum_{a_j \neq a} k_D(E(x), C_{a_j})) \Big], \tag{2}$$

where $k_D$ is some distance function defined over either a hidden representation of the discriminator $D$ or output probability $q_D(a|E(x))$ itself and $C_{a_j}$ is the moving average of the centroid for attribute $a_j$ in the attribute perception. Although there are many valid choices for $k_D(.,.)$, including the simple $l_2$ norm, we primarily use the well-known Kullback–Leibler (KL) divergence. We initialized $C_{a_j}^0$ via computing the centroids using all training data points.

The key of the proposal is that it only requires the same number of parameters as AFL, but it ensures that representations of an attribute approach to the representations of the other attributes. Also, it inherits the merit of AFL that leveraging the discriminative power to measure the invariance. Figure 2-(a–c) shows the behavior of the APM under the exact same experimental settings shown in Figure 1. The proposed method maximizes the conditional entropy significantly faster than AFL: $1.061$ after only five iterations, but AFL gives $0.683$ after $40$ iterations. Note that the value match the theoretical maximum value of the conditional entropy ($\log \frac{1}{3} \approx 1.061$). The comparison of the full plot of NLL (Figure 2-(d,e) also shows the faster convergence and stable behavior of the APM.

Another merit of the proposed method is that we can enforce semantic alignment with a simple modification. Individually, semantic alignment can be carried out by merely computing the centroids for each (attribute, label) tuple and aligning the perceptions of $\{x, y, a\}$ between only centroids of the same label $y' = y$ but different attributes $a' \neq a$. Although this modification does not help to maximize the conditional entropy, it prevents performance degradation of predicting $y$. Since most applications of invariant feature learning require to keep a information about $y$ we use this modification for all the later-described experiments.

## 4 EXPERIMENTS

We use three domain generalization tasks and two user-anonymization tasks. MNISTR (Ghifary et al., 2015) and PACS (Krizhevsky et al., 2012) is a well known image-based datasets designed for the purpose of domain generalization. We also test the methods of noise-robust speech recognition

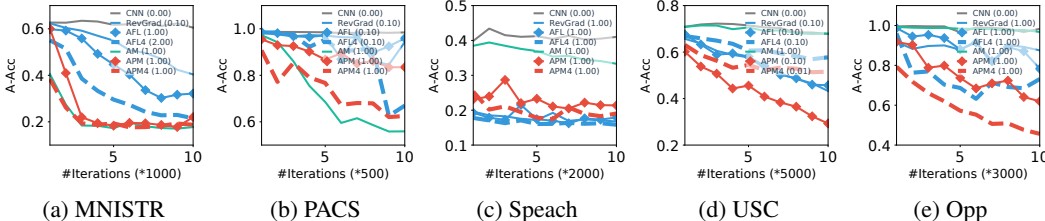

| (a) MNISTR | (b) PACS | (c) Speach | (d) USC | (e) Opp |

Figure 3: Performance comparisons of invariant representation learning. AFL4 and APM4 is a variant of the AFL and APM where the training steps of the discriminator $\kappa = 4$.

Table 1: Classification accuracies on unseen domains. We use $\kappa$=4 for AFL and APM on PACS.

| | MNISTR | | | | | | | Speech | | | | | PACS | | | | | All |
| | M0 | M15 | M30 | M45 | M60 | M75 | Avg | dishes | dishes+ | tap | tap+ | Avg | A | C | P | S | Avg | Avg |
|---|---|---|---|---|---|---|---|---|---|---|---|---|---|---|---|---|---|---|
| CNN | 82.7 | 99.1 | 96.9 | 91.3 | 97.8 | 88.5 | 92.72 | 86.7 | 83.3 | 86.7 | 81.7 | 84.61 | 57.1 | 61.9 | 80.9 | 58.7 | 64.66 | 80.66 |
| RevGrad | 85.6 | 98.8 | 97.6 | 90.6 | 96.3 | 87.5 | 92.73 | 86.2 | 83.8 | 88.2 | 81.9 | 85.01 | 57.0 | 61.1 | 81.9 | 58.7 | 64.65 | 80.80 |
| AFL | 85.2 | 98.0 | 97.5 | 92.0 | 95.9 | 87.3 | 92.64 | 86.8 | 83.6 | 87.6 | 81.8 | 84.93 | 59.0 | 61.0 | 83.9 | 60.5 | 66.11 | 81.23 |
| Crossrad | 84.3 | 98.9 | 97.7 | 92.7 | 98.2 | 88.0 | 93.30 | 86.2 | 84.8 | 87.8 | 82.8 | **85.42** | 57.5 | 62.1 | 81.4 | 63.6 | 66.17 | 81.63 |
| AM | 90.3 | 99.1 | 98.5 | 95.9 | 97.9 | 89.5 | **95.20** | 86.2 | 83.0 | 88.0 | 82.4 | 84.92 | 60.7 | 64.1 | 82.3 | 58.7 | **66.46** | **82.19** |
| APM | 90.4 | 98.6 | 98.2 | 94.2 | 98.0 | 87.9 | **94.57** | 88.7 | 84.0 | 87.3 | 83.9 | **85.98** | 62.5 | 63.7 | 83.4 | 59.5 | **67.27** | **82.61** |

scenarios using Google Speech Command Dataset (Speech). Regarding user-anonymization, we use two user-anonymization tasks on the data of wearables, OppG and USC (Iwasawa et al., 2017). The neural networks require to learn representations that help activity classification and at the same time, prevent to access the information about users (userID). As baselines, we use (1) A **CNN** trained on the aggregation of data from all source domains. (2) **AFL** (Xie et al., 2017), which was explained in Section 3.1. (3) **RevGrad** (Ganin et al., 2016) is a slightly modified version of AFL, which uses the gradient reversal layer to train all the networks. (4) **CrossGrad** (Shankar et al., 2018) is regarded as a state-of-the-art method in domain generalization tasks. (5) **Activation Matching (AM)**, which trains the encoder with the regularization of the $l_2$ distance on a feature space. (6) **APM** is our proposal. For all datasets and methods, we used RMSprop for each optimization. For all datasets except PACS, we set the learning rate to $0.001$ and the batch size to $128$. For PACS, we set the learning rate to $5e - 5$ and the batch size to $64$. For a fair comparison, hyperparameters were tuned on a validation set for each baseline. For the adversarial-training-based method, we optimized weighting parameter $\lambda$ from $\{0.001, 0.01, 0.1, 1.0\}$, except for MNISTR, for which it was optimized from $\{0.01, 0.1, 1.0, 2.0\}$. The value of $\alpha$ for CrossGrad was selected from $\{0.1, 0.25, 0.5, 0.75, 0.9\}$. We set the decay rate $\gamma$ to $0.7$ for all experiments. In all the experiments, we selected the data of one or several domains for the test set and used the data of a disjoint domain as the training/validation data. Specifically, we split the data of the disjoint domain into groupings of $80\%$ and $20\%$. We denote the test domain by a suffix (e.g., MNISTR-M0). We measured the label classification accuracy and the level of invariance. We measured the level of invariance by training a post-hoc classifier $f_{eva}$ following previous studies Xie et al. (2017); Iwasawa et al. (2017).

Figure 3 compares the performance the invariance of representations. For each method, we used the largest weighting parameter $\lambda$ on the condition that label classification accuracy did not significantly decrease. The results show that the proposed method stably achieves better invariant representations except for speech dataset. For example, APM achieved $20\%$ of the A-Acc in MNISTR, which is nearly perfect invariance as there are five domains in the validation data, while RevGrad and AFL achieved $30\%$ at best. On speech dataset, the performance of AFL and APM is mostly similar. These results confirm that the proposed method stably achieves more invariant representation compared with AFL. Table 1 summarizes the methods' classification performance on three different datasets: MNISTR, Speech, and PACS. The leftmost column of each table represents the test domain. We report the mean accuracy. We can make the following observations. (1) APM demonstrates the best or comparable performance on all three datasets, including CrossGrad, which is regarded as the state-of-the-art method in domain generalization tasks. (2) RevGrad and AFL often fail to improve performance even when compared with a standard CNN. These results suggest that the previous adversarial-training-based method suffered from the lack of semantic alignment when applied to domain generalization. (3) The Wilcoxon rank sum test shows that APM is statistically better than CNN, RevGrad, and AFL with $p < 0.01$, and than CrossGrad with $p < 0.05$.

## 5 CONCLUSION

This paper proposes a new approach to incorporating desired invariance to representations learning, based on the observations that the current state-of-the-art AFL has practical issues. Empirical results on both toy and real-world datasets support the stable performance of the proposed method to learn invariant features and superior performance on domain generalization tasks.

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

## A    PROOF OF THE THEOREM 1

*Proof.*  Using the Lagrange multiplier method, the derivative of

$$L = - \sum_{a \in \mathcal{A}} p(a, z) \log p(a|z) + \lambda(1 - \sum_{a \in \mathcal{A}} p(a|z)) \tag{3}$$

is equal to zero for the maximum entropy $H(A|Z)$. Solving the simultaneous equations, we can say $p(a{=}1|z) = p(a{=}2|z) = \cdots = p(a{=}K|z) = \frac{1}{K}$ for all $z \in Z$ when the conditional entropy is maximized, and based on the definition, the conditional entropy become $-\log \frac{1}{K}$.

From Bayes' law,

$$\frac{p(z|a = i)p(a = i)}{p(z)} = \frac{p(z|a = j)p(a = j)}{p(z)} \tag{4}$$

holds $\forall i \neq j \in A$ and $z \in Z$.                                               □

