# OpenReview forum: "Invariant Feature Learning by Attribute Perception Matching"
_ICLR.cc/2019/Workshop/LLD — LLD 2019_

### Official Review · AnonReviewer2 · 2019-04-09
**Improving adversarial feature learning by reformulating the conditionally entropy maximisation to a pair-wise distribution matching**

**Rating:** 4
**Confidence:** 1

**Review:**

The authors attempt to improve on Adversarial Feature Learning (AFL) by introducing a distrance between a parametrization of the attribute distribution and the Encoder Distribution. The resulting work appears to be easier to fit and delivers improved results compared to state of the art work.
The work seems relatively incremental, however, the results appear to justify the direction.
However, the exposition of the work is rather difficult to follow with notation popping without being previously defined, e.g. L_y section 3 last paragraph. I would strongly urge the authors to focus on clarifying the exposition of the work.

---

### Official Review · AnonReviewer1 · 2019-04-12
**Active enforcing of invariance instead of avoiding unwanted discriminability**

**Rating:** 3
**Confidence:** 1

**Review:**

This paper investigates Attribute Perception Matching as an alternative to Adversarial Feature Learning. While Adversarial Feature Learning attempts to avoid a discriminator being able to discern a certain set of labels that should not be inferrable from a learned representation, the proposed approach attempts to actively generate similarity between the representations learned, across the different unwanted labels. It is also shown that AFL might be unstable.

The example gives some intuition but may not exhibit the same behavior with actual high-dimensional data.

No notation is introduced in the paper, making especially the beginning very hard to follow.
On the other hand the entropy of a uniform distribution is elevated to Theorem status, and proven in the appendix, which is not necessary.

There is a typo in the heading of section 3. Overall the paper is quite hard to understand.

The contribution of APL merits publication, but it seems to this reviewer that it actually misses the topic of this workshop.

---

### Decision · Program_Chairs · 2019-04-16
**Acceptance Decision**

Accept